# The effect of virtual visual scene inclination transitions on gait modulation in healthy older versus young adults—A virtual reality study

**Amit Benady**[1,2,3,4], **Sean Zadik**[1], **Adi Lustig**[1,5], **Sharon Gilaie-Dotan**[2,6,7], **Meir Plotnik** [1,5,8]*

**1** Center of Advanced Technologies in Rehabilitation, Sheba Medical Center, Ramat Gan, Israel, **2** School of Optometry and Vision Science, Bar Ilan University, Ramat Gan, Israel, **3** Orthopedic Department, Tel Aviv Sourasky Medical Center, Tel Aviv, Israel, **4** Anatomy and Anthropology Department, School of Medicine, Tel Aviv University, Tel Aviv, Israel, **5** Department of Physiology and Pharmacology, Faculty of Medicine, Tel Aviv University, Tel Aviv, Israel, **6** UCL Institute of Cognitive Neuroscience, London, United Kingdom, **7** The Gonda Multidisciplinary Brain Research Center, Bar Ilan University, Ramat Gan, Israel, **8** The Sagol School of Neuroscience, Tel Aviv University, Tel Aviv, Israel

* meir.plotnikpeleg@sheba.health.gov.il

**Data Availability Statement:** We uploaded our data set to the Figshare public repository- https://doi.org/10.6084/m9.figshare.27129879.

## Abstract

Bipedal locomotion requires body adaptation to maintain stability after encountering a transition to incline walking. A major part of this adaptation is reflected by adjusting walking speed. When transitioning to uphill walking, people exert more energy to counteract gravitational forces pulling them backward, while when transitioning to downhill walking people break to avoid uncontrolled acceleration. These behaviors are affected by body-based (*proprioception* and *vestibular*) cues as well as by *visual* cues. Since older age adversely affects walking, it is unclear whether older adults rely on vision during locomotion in a similar manner to younger individuals. In this study, we tested whether the influence of visual cues on these walking speed modulations in healthy older adults (60–75 years old, N = 15) were comparable to those found in healthy young adults (20–40 years old, N = 12). Using a fully immersive virtual-reality system embedded with a self-paced treadmill and projected visual scene, we manipulated the inclinations of both the treadmill and the visual scene in an independent manner, and measured participants walking speed. In addition, we also measured individual visual field dependency using the rod and frame test. The older adults presented the expected braking (decelerating) and exertion (accelerating) effects, in response to downhill and uphill visual illusions, respectively, in a similar manner to the young group. Furthermore, we found a significant correlation between the magnitude of walking speed modulation and visual field dependency in each of the groups with significantly higher visual field dependency in older adults. These results suggest that with aging individuals maintain their reliance on the visual system to modulate their gait in accordance with surface inclination in a manner similar to that of younger adults.

**Funding:** This study was supported in part by the Israel Science Foundation (ISF) grant #1657-16 awarded to MP. SG-D was supported by the Israel Science Foundation (ISF)- grant #1485-18 URL: https://www.isf.org.il/#/ The funder did not play any role in the study design, data collection and analysis, decision to publish, or preparation of the manuscript.

**Competing interests:** The authors have declared that no competing interests exist

## Introduction

Nowadays, with the increase in life expectancy and with older adults becoming a larger portion of humanity, there is growing interest in the manifestation of aging in different behavioral domains. Locomotion is one of the behaviors that changes significantly with age. To control locomotion the motor system utilizes Multi-Sensory Integration (MSI), i.e., combining multiple unisensory inputs into a unique coherent motor plan. The three main sensory inputs are the *proprioception* and *vestibular* cues that are known together as body-based cues, and the *visual* cues [1–3] where usually all three are integrated in a synergia to maintain stable locomotion. To evaluate the contribution of visual cues as part of the MSI, there is a need to artificially manipulate them in an independent fashion from the body-based cues and explore their relative "weight". To that end, we developed a novel paradigm using a fully immersive virtual-reality (VR) system where participants walk on a treadmill operated in a self-paced mode and are presented with virtual visual scenery projected on a large dome-shaped screen [4–7]. During walking trials, the visual scene's apparent inclination is transitioned independently of the physical inclination of the treadmill allowing us to examine gait modulations following these transitions. Previous studies in our lab showed that in young healthy adults walking on a leveled treadmill, uphill virtually visually simulated transition is followed by a temporary increase in gait speed, while downhill virtually visually simulated transition is followed by a temporary decrease in gait speed [4–6]. These visually guided brief gait speed modulations represent the virtually induced *exertion* and *braking* effects that are typically evident during physical uphill and downhill walking, respectively. In addition to independently manipulating visual cues during locomotion, we also examined individual's visual field dependence (considered as the level of reliance on visual cues in comparison to body-based cues [8, 9]) by the commonly used rod and frame test [6, 10–12] and found that gait speed modulations were associated with visual field dependency [6]. However, these results were obtained for young healthy adults and it is unclear whether they generalize to older adults. In the older adult population, many aspects related to locomotion change. Balance control, the ability to maintain stability (both *stable posture* referring to static stability and *stable locomotion* referring to dynamic stability), especially while propelling the body forward, is an important aspect of locomotion that was reported to become reduced in older adults [13]. This reduction is due to several changes that occur with aging and involves alterations in the MSI component weights i.e., delayed and/or decreased vestibular and somatosensory influx, which in turn increase visual dependency with aging [14–16]. Interestingly, this compensatory increase of visual dependency in the MSI takes place even though visual acuity is known to deteriorate with age [17]. An additional major age-related change is the decrease in gait speed. Gait speed is known to be the most prominent age-related change with spontaneous gait speed decrease of about 1% per year from the age of sixty and afterwards [13, 18, 19]. In addition, visual field dependency varies across healthy people [20, 21] and was found to be higher in individuals with pathologic or physiologic (e.g., aging) balance-related conditions [22–26].

In the present study we turned to explore to what extent the effects we found in younger adults [6, 7] are preserved in older adults. Since there is growing reliance on vision with aging [27, 28], we hypothesized to see similar locomotor adaptations following virtually visual transitions in an elderly healthy population, but with a higher relative change in walking speed in comparison to the young healthy population. Furthermore, we hypothesized that the visual field dependence would be higher in the elderly population yet associated to their gait speed modulations.

## Materials and methods

### Participants

For this study we collected data from fifteen healthy older adults (mean age ± SD: 68.93±4.17 years old, twelve males) and used existing data from previous study [6] of twelve young healthy participants (mean age ± SD: 26.53±3.09 years old, six males). The recruitment period for this study spanned from the ninth of July 2020 to the twenty fifth of May 2021. Exclusion criteria were physical or visual restrictions, cognitive limitations, and any sensorimotor impairments that could potentially affect locomotion. Specifically, participants were required to self-report having vision acuity 20/20 (or corrected to 20/20 by means of glasses/contact lenses) and no color blindness. Additionally, participants were inquired on any neurological, vestibular or orthopedic conditions that may hinder their ability to walk, in addition to any other movement restrictions. The IRB at the Sheba Medical Center, Israel, approved the experimental protocol, and all participants signed a written informed consent prior to entering the study.

### Apparatus

**Virtual reality system.** Walking experiments were conducted in a fully immersive virtual reality system (CAREN High End, Motek Medical, The Netherlands) containing a moveable platform with six degrees of freedom. A self-paced treadmill embedded in the movable platform allowed participants to adjust the treadmill speed according to their preferred walking speed [29].

### Procedure

**VR rod-and-frame test.** The rod-and-frame test was the first task participants were engaged with in the experiment (after filling the informed consent). Firstly, we made sure that the participant felt comfortable with the head mount device (HMD) by which the task was administered (see Section A in S1 File). Secondly, the participants underwent a short practice trial to confirm that they fully understood the task. Afterwards, 28 test trials were conducted and measured accurately. There was no time limit, and typically the whole session lasted 10 minutes, including the practice trial.

**Gait trials in a large-scale VR system.** First, participants walked in self-paced mode in leveled and inclined surfaces during a short habituation period. The participants were attached to a safety harness on the moveable platform during all walking conditions (Fig 1). The first part of the habituation took about 10–15 minutes and was aimed to familiarize the participants with the self-paced mode of the treadmill. During this part, the participants walked at their preferred speed until they felt comfortable, then the experimenter asked them to increase and decrease their speed until they mastered the walking in a self-paced mode. The second part of the habituation included one walking trial of each of the three possible inclinations (i.e., level, uphill, and downhill) when the visual and the physical cues were synchronized ('congruent' conditions; see more details below). Each trial lasted three to four minutes.

After the habituation period, participants were introduced to the experimental conditions in random order. Participants were informed that they would perform several short gait trials with short intervals between them. They were instructed to walk "as naturally as possible" and that "inclinations may occur during walking". Each walking condition started with the participant in a standstill position and then progressed into walking with both the treadmill and the visual scene leveled until reaching steady-state velocity (SSV) for 12 seconds. Detailed explanations on how we defined SSV are provided in the Section B in S1 File. Once reaching SSV, a 5-second-long transition of the treadmill and/or visual scene

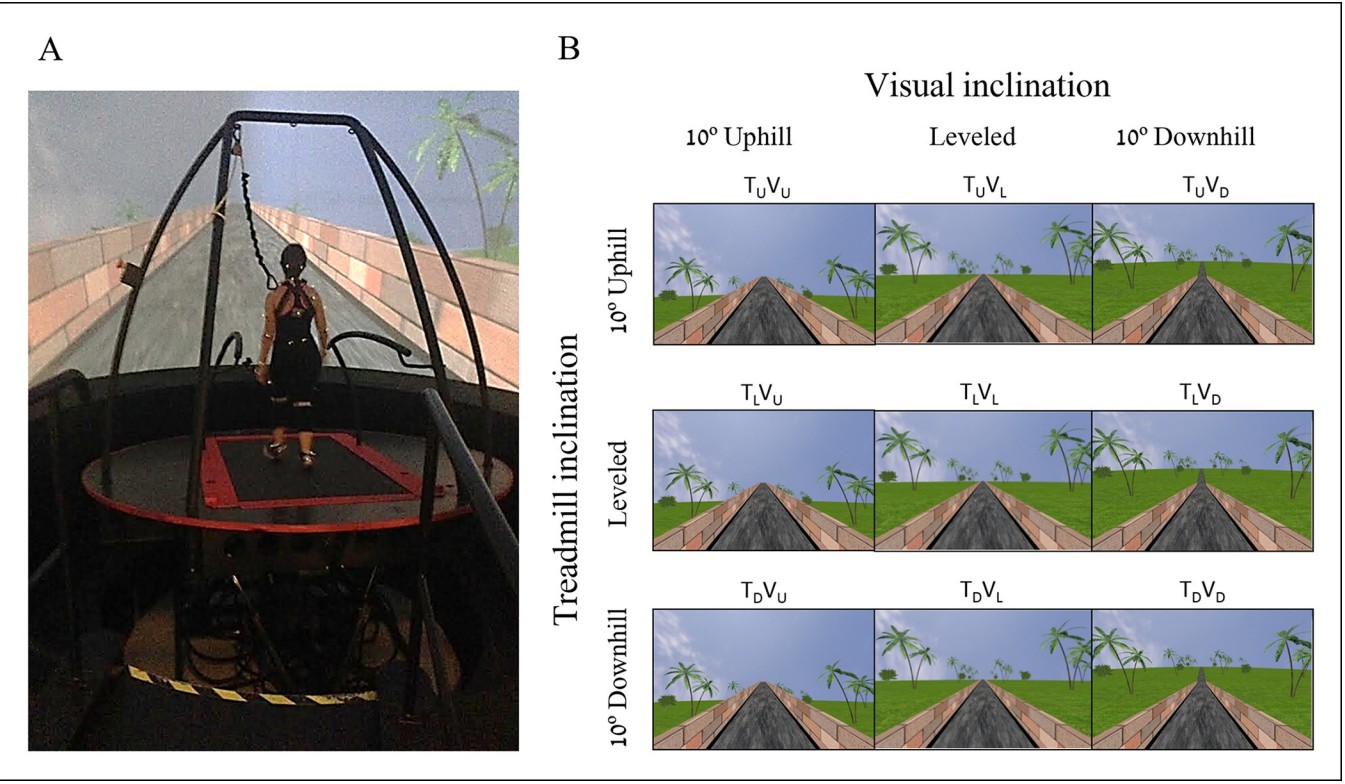

**Fig 1. Apparatus and experimental conditions.** (A) A fully immersive virtual reality system containing an embedded treadmill synchronized with projected visual scenes, wherein this example, the treadmill is leveled, and the vision is uphill (see $T_LV_U$ in B). (B) Experimental conditions. Each condition started with leveled walking and only after reaching steady-state velocity (SSV), a transition (5s) occurred to one of nine walking conditions presented in a random order. The conditions include transitions of the treadmill (T) and/or visual scenes (V) to 10° uphill (U), remaining leveled (L) or to -10° downhill (D). Rows depict treadmill inclination changes, and columns depict visual scene inclination changes. Visual scenes inclination effect was achieved by the road appearing above (uphill), below (downhill), or converging (leveled) with the line of the horizon. In addition, the peripheral greenery is exposed more (downhill) or less (uphill) by the road. Vision-treadmill congruent conditions appear on the diagonal ($T_LV_L$ for continued leveled walking, $T_UV_U$ and $T_DV_D$ for uphill and downhill walking, respectively). Vision-treadmill incongruent conditions below the diagonal represent conditions with visual scene inclination more positive than the treadmill's ($T_LV_U$, $T_DV_U$, $T_DV_L$), and above the diagonal visual scene inclination more negative than the treadmill's ($T_UV_L$, $T_UV_D$, $T_LV_D$). See Methods for more details. (Adopted from Benady et al., 2021 [6]).

occurred (except in the congruent leveled condition, where no actual transition occurred). Post transition, each condition lasted another 65 seconds, until the treadmill slowed down and stopped altogether.

**Experimental conditions.** The protocol included nine walking conditions that the participant encountered in random order. Inclination of the treadmill (T) and/or visual scenes (V) transitioned to 10° uphill (U), remained leveled at 0° (L) or transitioned to -10° downhill (D). Fig 1 shows the 3x3 experimental design, where rows represent treadmill (T) inclination and columns represent visual scene (V) inclination. Congruent conditions where the treadmill and visual scene inclinations are synchronized were set as baselines. These conditions appear on the diagonal of Fig 1; upper left- uphill ($T_UV_U$), middle- leveled ($T_LV_L$) and bottom right- downhill ($T_DV_D$) walking conditions. Treadmill-vision incongruent conditions include the following visual scene manipulations: for treadmill uphill inclination, the vision was leveled ($T_UV_L$) or downhill ($T_UV_D$), for treadmill leveled inclination, the vision was uphill ($T_LV_U$) or downhill ($T_LV_D$), and lastly, for treadmill downhill inclination, the vision was leveled ($T_DV_L$) or uphill ($T_DV_U$).

## Outcome measures

**Gait speed related variables.** To assess the post-transition effects on gait speed, we examined (i) the magnitude of the peak/trough of gait speed relative to the SSV (%); and (ii) the time of this peak/trough after the transition (seconds). We refer to the transition start time as time zero (t = 0).

Additionally, we normalized gait speed for each participant in each experimental condition, the process consisted of three steps as follows: (i) gait speed was divided by the averaged SSV at every second, (ii) the ratio between gait speed and SSV was presented as a percentage, and (iii) the normalized trace was shifted so that the mean value of the SSV period would be zero. Following these steps, it was clear to distinguish between the responses of increased and decreased speed following the transition.

**Standardized response to virtual inclination.** To compute this index, we used data from the incongruent $T_LV_U$, $T_LV_D$, $T_DV_U$, $T_UV_D$ walking conditions. The averaged absolute values from these four conditions of the peaks/troughs (%) relative to the SSV were calculated for each participant.

**Visual field dependence index.** In each trial of the rod and frame test, the degree of deviation of the rod from the true upright position was recorded as the position error. For each participant, the mean position error of the seven different frame angles was calculated. Data from all participants were grouped by the frame angle [20]. We defined the visual field dependence index as the average position error when the frame was projected at ±20 degrees (8 trials in total, 4 trials of +20˚ and 4 trials of -20˚). This parameter allowed us to evaluate individual differences in visual field dependence.

**Ratio of gravity-induced behavior.** To assess the level of influence of gravity on walking speed (WS), we calculated the area under the curve (AUC) of both the velocity of a free body over time (second-by-second i.e., at 1 s, at 2 s... at 60 s), that is, $V(t) = g*sin\theta*t$ (where g is the gravitational constant and $\theta$ is the inclination angle) and of WS for congruent uphill and downhill conditions. We defined the ratio according to the following equation (we added a multiplier of 100 to avoid extremely small numbers):

$$R = [(AUC(WS_i))/(AUC(V(t)_{t=i}))]*100 \tag{1}$$

The index "i" refers to the time (in seconds) post-transition. The ratio quantifies the degree to which WS approximates the velocity of a free body. A positive ratio indicates that both WS and a free body accelerate/decelerate in the same direction (i.e., both accelerate when downhill and decelerate when uphill). A negative ratio indicates that WS acceleration was in the opposite direction from this of a free body. A ratio further from zero suggests more gravitational influence on walking.

**Linear summation model.** Given that locomotion is maintained by MSI, we used a linear weighted summation model [1] to estimate the weight of visual and body-based cues [4]. The general model is presented by the following equation:

$$WS_{combined} = W_{visual\ cues}*WS_{vision\ driven} + W_{body-based\ cues}*WS_{body-based\ driven} \tag{2}$$

$WS_{combined}$ is defined as the resulting integrated behavior (walking speed) after summing the relative part of each component. On the right side of the above equation, W denotes the weight of a unimodal cue (*visual* or *body-based*) and WS refers to the walking speed which was driven solely by that cue (*visual* or *body-based*). We assume that WS in conditions $T_LV_U$ ($WS_{vision\ driven,\ up}$) and $T_LV_D$ ($WS_{vision\ driven,\ down}$) are driven exclusively by uphill and downhill visual cues, respectively. Contrariwise, we assume that WS in conditions $T_UV_L$ ($WS_{body-based\ driven,\ up}$) and $T_DV_L$ (WS $WS_{body-based\ driven,\ down}$) are driven exclusively by the

corresponding body-based cues. While assuming that vision and body-based cues are the main factors influencing locomotion, the weight of each unimodal cue can be calculated using the following general equations:

$$W_{visual\ cues} + W_{body-based\ cues} = 1 \tag{3}$$

$$W_{visual\ cues} = \frac{(WS_{combined} - WS_{body-based\ driven})}{(WS_{vision\ driven} - WS_{body-based\ driven})} \tag{4}$$

$$W_{body-based\ cues} = \frac{(WS_{combined} - WS_{vision\ driven})}{(WS_{body-based\ driven} - WS_{vision\ driven})} \tag{5}$$

recall $WS_{combined}$ represents the congruent conditions $T_U V_U$ and $T_D V_D$ (i.e., combined cues) and can be calculated using the following equations for each respective inclination:

$$WS_{conbined,up} = W_{vision\ cues,up} * WS_{vision\ driven,up} + W_{body-based\ cues,up} * WS_{body-based\ driven,up} \tag{6}$$

$$WS_{conbined,down} = W_{vision\ cues,down} * WS_{vision\ driven,down} + W_{body-based\ cues,down} * WS_{body-based\ driven,down} \tag{7}$$

Notably, to avoid repetition of mathematical factors ('using data to explain related data'), the predicted behavior of uphill and downhill walking was estimated using the opposite unimodal weights i.e., both visual and body-based weights derived from downhill conditions was used for the prediction of uphill walking, and the uphill derived weights was used for downhill walking.

## Statistical analyses

Differences in the magnitude and timing of normalized walking speed in the incongruent walking conditions (i.e., $T_L V_U$ and $T_L V_D$) between the young and older adult groups were compared at the peak/trough of change. In pre-hoc analyses, Shapiro-Wilk normality tests were conducted on the time and amplitude of peak outcome measures (in both $T_L V_U$ and $T_L V_D$ conditions), for each age group (n = 2; young and older adults). Of the eight data sets assessed, normality tests indicated partially non-normal distributions for the older adults group. Accordingly, a non-parametric approach was chosen for the post-hoc age group comparison using the Mann-Whitney test. We than confirmed all outcome measures sphericity (i.e., equal variances of the differences between all groups) using the Levene's test (p>0.05). A p-value of equal to or less than 0.025 (considering two variables) indicated a significant difference and results effect size was estimated using the rank biserial correlation ($r_{rb}$), in accordance with the non-parametric approach. The difference between the rod and frame index of both groups was compared with the same method. Additionally, we calculated and compared the time it took the participant to return from the peak/trough velocity to SSV. Spearman correlation coefficient was computed to evaluate the association between visual field dependence index and (i) the standardized response to virtual inclination and (ii) the age of participants. Data were analyzed using SPSS software (v25, IBM).

## Results

### Gait speed modulations following physical and/or virtual inclination transitions

We initially compared the behavioral change measured by gait speed modulations in the older versus young adult's groups following a virtual (i.e., visual) and/or physical (i.e., treadmill) inclination transitions. Fig 2 depicts the normalized average of walking speed relative to the SSV for each condition, one minute pre- and post-transition. For the treadmill leveled conditions, the same patterns were seen between the groups for all three visual scene inclinations. Specifically, when the visual scene remained leveled ($T_L V_L$), both groups maintained roughly the same speed as the SSV. When the visual scene transitioned upwards ($T_L V_U$- i.e., virtually induced exertion effect), both groups temporarily increased their walking speed until reaching a peak. No significant change was seen either in the time of the peak (mean±SE: young = 10.37 ±0.64sec, older adults = 11.05±1.34sec, $p > 0.99$, $r_{rb}$ = -0.006), nor in the amplitude of the peak (young = 18±3%, older adults = 12±4%, $p$ = 0.43, $r_{rb}$ = -0.183). Note that after the peak, the older adult group returned to their SSV, while the young group decreased their speed until a new steady state, which was higher than their previous SSV. Regarding the opposite condition,

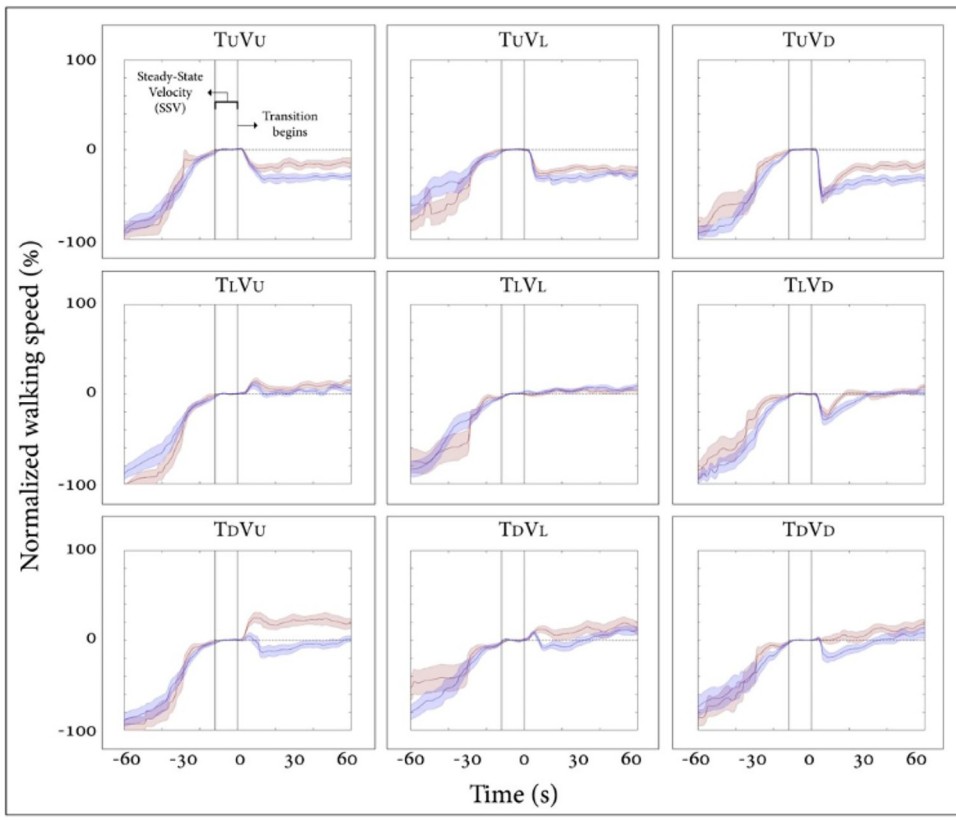

**Fig 2. Normalized walking speed in the young (N = 12) and older adults (N = 15) groups.** X-axis represents the time (seconds); 60 seconds before and after the transition, y-axis represents the normalized average self-paced gait speed relative to steady-state velocity (%) for each condition. Orange lines represent the young group and purple lines the older adult group, shaded colors represent the standard error for each group. Time zero demarcates the end of the steady-state velocity period, after which a 5s transition of the treadmill and/or visual scene occurred. Conditions: treadmill (T) and/or visual scenes (V) inclination transitioned to 10˚ uphill (U), remained leveled at 0˚ (L), or transitioned to -10˚ downhill (D). Note that there was no difference in speed modulation patterns or magnitude between both groups following the visually induced *braking* and *exertion* effects.

where the treadmill remained leveled, but the visual scene transitioned downward ($T_L V_D$- i.e., virtually induced braking effect), both groups decreased their speed until reaching a trough with no significant difference in the peak's time (young = 8.08±0.59sec, older adults = 8.06 ±0.48sec, $p = 0.98$, $r_{rb} = 0.011$), nor between the magnitude of both groups (young = 25±5%, older adults = 32±6%, $p = 0.63$, $r_{rb} = 0.111$). After the trough, the young group returned to their previous SSV after 18.94±4.16sec while the older adult group returned to their SSV after 26.23±2.69sec ($p = 0.045$, $r_{rb} = 0.46$).

For the treadmill up conditions there was a gradual monotonic change in speed in both groups, with a more prominent effect in the older adult group (see Fig 2; first row, purple line denoting the older adult group, is lower i.e., slower than the orange line representing the young group). Interestingly, in the treadmill down conditions both groups showed opposite walking patterns; in the congruent condition $T_D V_D$, while the young group constantly increased their walking speed, the older adult group initially braked and decreased their speed, with a monotonic increase during the next minute after the transition. This pattern was roughly the same also in the $T_D V_U$ and the $T_D V_L$ conditions.

## Relation between visually induced modulation of gait speed during visual-physical incongruent conditions and visual field dependence

To further test our findings from our previous study [6] that showed a significant correlation between the magnitude of visual modulation on gait speed during virtual inclination changes and individual's visual field dependency in a young group, we now computed these measures for the older adult group. We calculated for each participant (i) the magnitude of visual modulation on gait speed based on the gait speed changes in the incongruent conditions (i.e., standardized response to virtual inclination, see Methods for calculation), and (ii) the index of visual field dependence estimated by the rod and frame test. As expected, we found also in the older adult group a significant correlation between the standardized response to virtual inclination and the index of visual field dependence estimated by the rod and frame test (Fig 3, upper panel; blue circles, Spearman's r = 0.63, $p = 0.012$). Data of young healthy adults were plotted on the same graph for comparison, also reflecting significant correlation between these two measures (Fig 3, upper panel; orange circles, Spearman's r = 0.79, $p = 0.004$). Furthermore, a significant correlation was seen between the rod and frame index and the age of the participant only in the older adult group (Fig 3, bottom panel; Spearman's r = 0.63, p = 0.013), but not in the young group. In addition, the Man-Whitney test showed significant difference ($p = 0.007$, $r_{rb} = 0.6$) between the rod and frame indices of the two groups (older versus young adults).

## Ratio of gravity-induced behavior in congruent uphill and downhill walking

To further compare between the two groups, we measured the braking and exertion effects post-transition over time. For this, we computed the normalized ratio between the areas under the curve of gait speed for the congruent uphill and downhill walking conditions, divided by free body velocity at the same inclination (i.e., $V(t) = g * sin \pm 10 * t$; where g is the gravitational constant; Fig 4). The higher the ratio (i.e., further from zero), the stronger the effect of gravity and the weaker the exertion and braking effects are. The analysis revealed a differential response to gravity forces between both groups, both in the uphill and downhill walking (see statistical results in figure legend). For uphill walking, the same pattern was seen between both groups (c.f. Fig 2, upper left corner), but with a greater ratio in the older adult group, reflecting a greater influence of gravity with a smaller exertion effect. The turning point, which

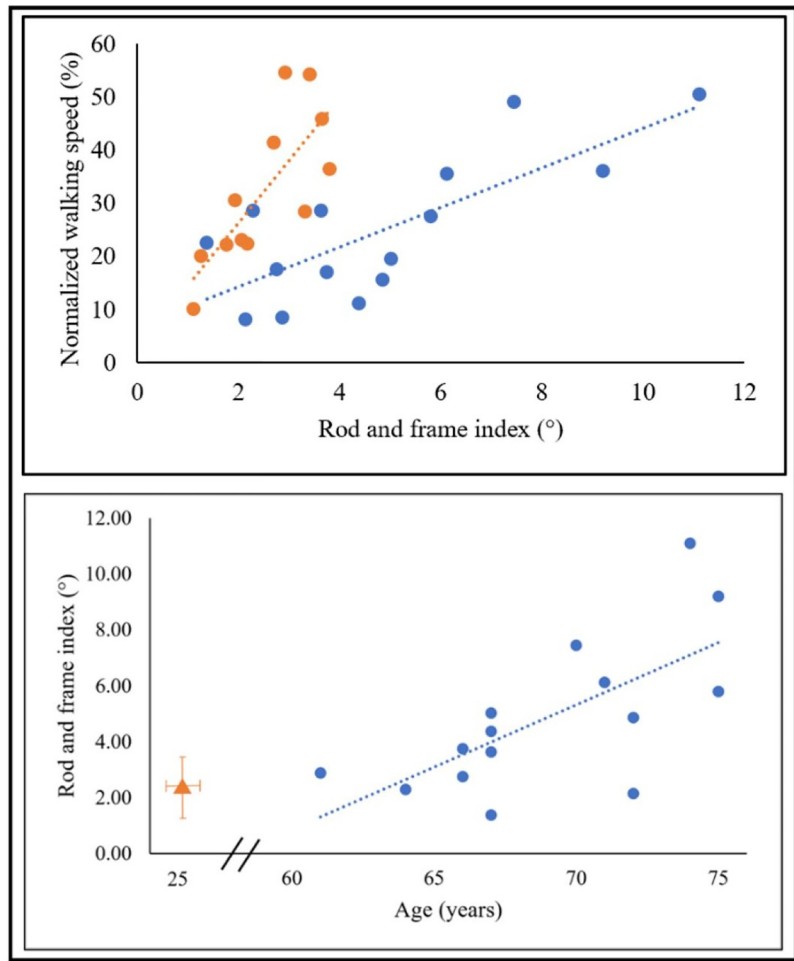

**Fig 3. Visual field dependence analyses.** Upper panel: Average standardized response to virtual inclination relative to the visual field dependence index for the older adult group (blue circles; N = 15) and young group (orange circles; N = 12). Each circle stands for one participant. X-axis represents the visual field dependence as assessed by the psychophysical rod and frame test while seated. Y-axis represents the standardized response to virtual inclinations based on treadmill-visual incongruent walking conditions. A significant correlation was seen between these two values for each group independently (young: Spearman's r = 0.79, $p$ = 0.004, older adults: Spearman's r = 0.63, p = 0.012). Bottom panel: Visual field dependence index relative to the participant's age. The extent of visual field dependency during locomotion in incongruent conditions is correlated with the age of the participant in the older adult group, but not in the young's group. X-axis represents the age of the participant (years), y-axis represents the rod and frame index. Blue circles represent the older adult group (N = 15), and the orange triangle represents the average of the young group (N = 12) for reference. A significant correlation was found in the older adult group (r = 0.63, $p$ = 0.013).

demarcates the time (in seconds after the transition) when the exertion effect begins, and participants start to counteract gravity forces, was similar in both groups ($T_{Uphill}$ = 10s). For downhill walking, the older adults group showed initially the same pattern of a monotonic decrease in walking speed, with eventually a gradual return to SSV. Their turning point was almost the same as in the uphill condition ($T_{Downhill}$ = 11s), meaning that in both inclinations, after 10-11s the older adults group applied the exertion/breaking effect to counteract gravity. The walking pattern for the young group was initially maintaining SSV with a gradual increase in speed. Their turning point was at 9s, in this case reflecting a decrease in the braking effect which prevents an uncontrolled acceleration forward, followed by a gradual monotonic increase of the walking speed (c.f. Fig 2, right bottom corner).

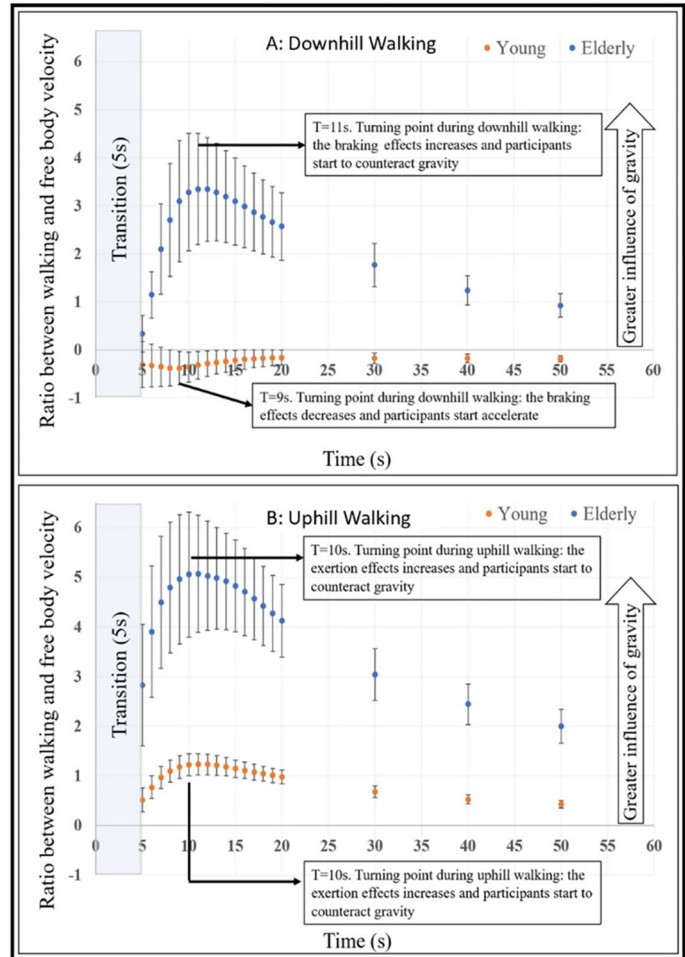

**Fig 4. Ratio of gravity-induced behavior in congruent uphill and downhill walking.** The average ratio between participants walking speed (WS) and velocity of a free-moving body (V(t)) over time for ±10˚ downhill (upper panel) and uphill (bottom panel) walking. Error bars represent SE. The turning points represent the time (seconds) in which the participants either applied or rejected the exertion or braking effects, to counteract or engage gravitational forces during walking. A statistical difference ($p<0.05$, corrected for multiple comparisons K = 19; number of calculated time points) was found between the young and older adults only after the turning points.

## Application of the linear summation model for congruent uphill walking in the older versus young adult groups

We applied the linear summation model (see Methods section) to evaluate the weight of *visual* cues in comparison to *body-based* cues over time during uphill walking (Fig 5, upper panel), and to further examine if our prediction indeed fits the true behavior during congruent uphill walking (Fig 5, bottom panel). This model assumes that true uphill walking is the integration of the unimodals contributing to it, in our case visual (vision-up) cues and body-based (tread-mill-up) cues. Each cue has a relative contribution for every time-period, but at any given time the sum of all unimodals equals one. This model enables estimation of the sensory reweighting of the MSI components. For example, a weight of body-based cues near 0 indicates that at the same time-point, locomotion predominantly relies on vision. Our model shows the relative weight of each unimodal cue as we continue walking. This model did not fit our participants for downhill walking as it is known that in young people walking modulations are less

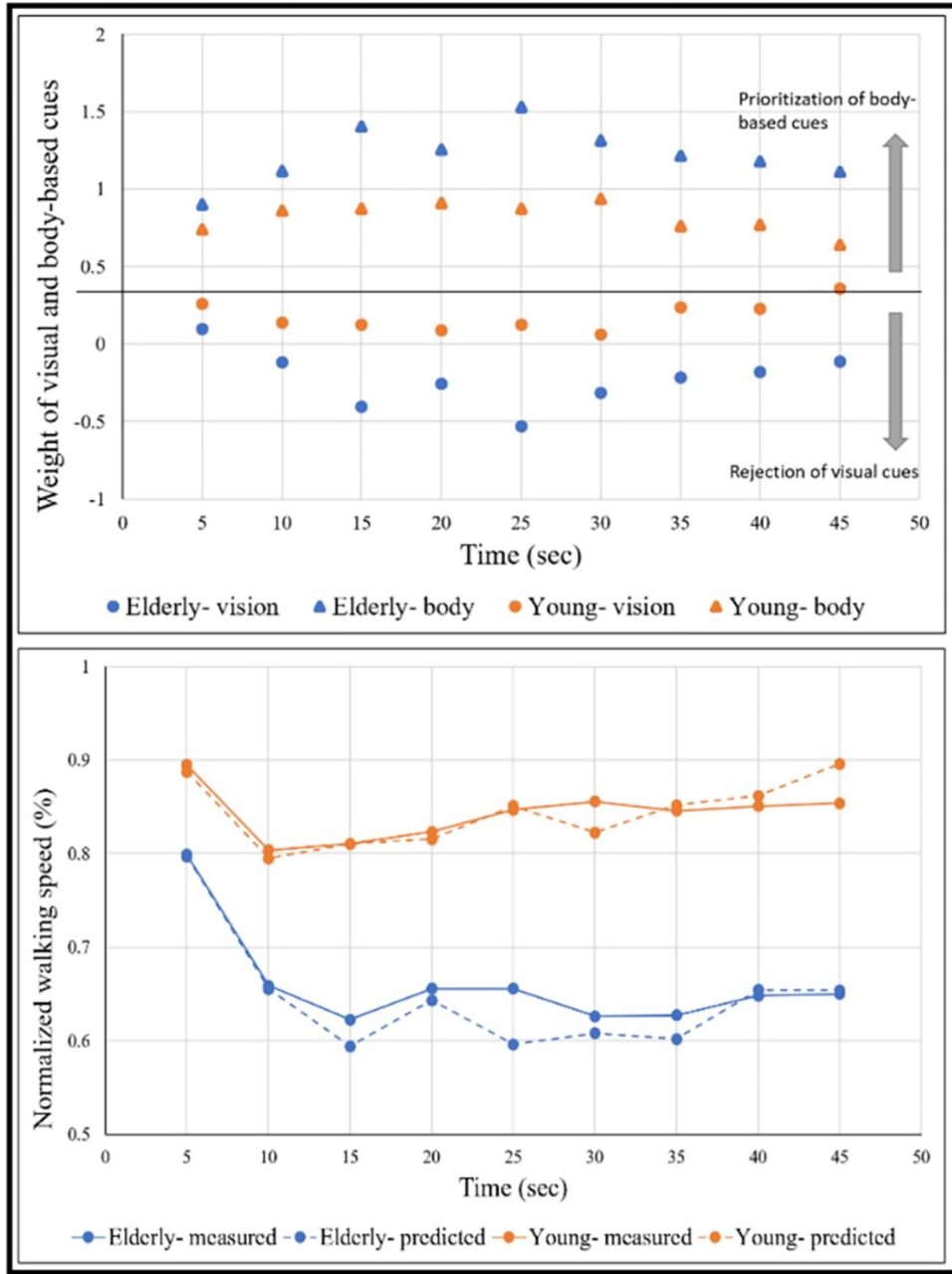

**Fig 5. Congruent uphill walking.** Upper panel: Sensory reweighting- the weight of visual and body-based cues (proprioception and vestibular) during locomotion over time. X-axis depicts the time (sec) with time zero demarcates the beginning of the transition (5s long). Blue color represents the older adult's group and orange color represents the young group. Circles represent the weight of visual cues; triangles represent the weight of body-based cues. Bottom panel: Linear summation model (see Methods). Comparing our measured gait speed to the expected gait speed according to the linear summation model over time in congruent uphill walking (c.f. Fig 2, upper left corner). Solid lines represent the measured walking speed and dashed lines represent the expected gait speed.

homogeneous during downhill walking as there are people who increase and people who decrease their speed [30–32], while older adults usually decrease their speed [33]. In contrast, uphill walking has a more consistent pattern with most people decreasing their speed, both in the young and older adult population [33, 34].

## Discussion

In this study, we compared the weight of visual cues on locomotion modulation under physical and/or visual changes between healthy young and older adults. We manipulated vision independently of body-based (proprioception and vestibular) cues and measured the changes in walking speed. We found that roughly the same pattern of walking adaptation to virtually induced braking and exertion effects (either slowing down or speeding up, respectively) was employed by both groups. We also observed a significant correlation between the magnitude of the virtually induced braking and exertion effects, and the individual's visual field dependency as measured by the rod and frame test for each group separately, but with significantly bigger reliance on visual field in the older adult group. Finally, we applied the linear summation model during uphill walking, which showed the relative weight of each unimodal cue as we continue walking in both the young and older adult groups and found the expected speed pattern fits the real walking speed pattern as measured in the congruent uphill condition. For further discussion regarding double incongruent conditions see Section C in S1 File.

### The relative weight of visual cues during physical and/or visual manipulations

**Treadmill leveled conditions.** When walking with the treadmill leveled, both groups showed the same pattern of gait speed modulation for all three virtual visual inclinations (c.f. Fig 2, middle row). As older adults have increased reliance on visual cues [27, 28], an observation that was confirmed in our study as can be seen by the results of the rod and frame test (Fig 3), we expected to see larger speed modulations following virtually induced braking and exertion effect. Surprisingly, no significant difference was seen in the magnitude nor the timing of the peaks/troughs. Lack in differences might be related to the cohort of elderly adults who participated in the study. While being above the age of 65, most of the participants were relatively fit, possibly reflected by the fact that they consented to participate in the study after being informed that a 10-degree inclined walking may be required. Thus, our study might be limited by the fact that the older adults of our study might not be representative for a typical ~70-year-old individual. We did not encounter young adults declining to walk at 10 degrees inclinations. This finding strengthens the notion that MSI deterioration is not affected primarily by aging, but rather by physical condition and lifestyle, factors that can be controlled and modified. At the same time, our study convincingly observed that while maintaining the behavioral constructs of braking and exertion effects, older adults resist the consequences of the gravitational forces upon their walking speed to a lesser extent than young adults do. This fact was reflected by the ratio of gravity-induced behavior. We believe that this finding is reflecting the general reduction in volition of locomotion that is also expressed by additional age-related modulations of speed walking [13, 18, 19].

### Inter-participant variability

In the healthy population, there is a well-established relationship between subjective visual vertical (SVV), which is thought to indicate visual field dependency [10, 12], and postural stability [22, 35]. However, the locomotive reactions, which are behaviorally reflected by variations in gait speed, are not fully understood. Herein we strengthen the notion that visual field dependency increases with aging [27, 28], this can be seen by (1) significant increase of visual field dependency as reflected by the 'road and frame' test index, in the older adults group compared to the young group, and (2) only within the older adults, there is a significant positive correlation between age and rod and frame index. The correlation between age and rod and frame

index strengthens our previous findings [6, 7] that visual dependence as measured independently by two orthogonal tests, i.e., subjective spatial computerized test, and behavior- based gait speed adaptations reflect the fact that assessment of gravity direction (rod and frame) and consequences (speed modulations) are interrelated in the relevant neuronal pathways. We posit that this finding may hold translational significance, for example, as a new evaluation approach that combines a short walking trial in a visual conflict paradigm with the rod and frame test can potentially estimate visual dependency in locomotion. This may aid in the identification of those who may benefit from visual conflict paradigms, e.g., for gait rehabilitation purposes allowing for a more individualized rehabilitation approach.

## Limitations

There are several limitations to our study. First, our study included a relatively small sample size, this is more profound within the older adults, as the variation across individuals is higher. Second, using a VR platform may not reflect real life behavior, even more in the older adult group which are less familiar to novel technologies and may need a longer habituation period to feel comfortable with the paradigm. Third, transitions to inclined walking were presented swiftly and within a time window of 5 seconds, while in real life, the individual can anticipate the inclination in advance, based on visual inputs from the environment. This could affect more the older adults, which have decreased vision acuity, and they might encounter the transition at a later stage [36]. Despite this limitation, the results of the present paradigm clearly discern between the roles of vision and body-based cues in adjusting gait behavior with reference to inclinations.

## Future directions

Overall, senses and body functions are known to deteriorate with aging, such as vision [36], joint mobility [37], muscle force [38] and balance [16]. However, age-related changes of multi-sensory integration are less studied. The changes in performance of MSI tasks may be more sensitive and aid as an earlier predictor to detect future deterioration in daily living activities compared to assessing each sense by itself. Taking a more holistic approach in which you evaluate the integrated outcome of sensorimotor function can be potentially beneficial in the clinical arena where diagnostic tools and early markers could be developed based on performance of daily MSI tasks.

Future studies should investigate among other aspects, also the effects of aging on MSI in relation to the autonomic nervous system (ANS) (e.g., heart rate variability). Normal aging is accompanied by a series of complex alterations in the autonomous control of the cardiovascular system, with an increase in the cardiac sympathetic nervous system tone and withdrawal in the parasympathetic nervous system. Although paradigms of incongruent feedback information provide some insight about the mediating effects of external performance cues on perceptual experience, very little is known about the mechanism that controls such behavior, and how it is manifested in the activity of the ANS, especially in older adults. Better understanding of these mechanisms can further improve detection and clinical treatment of the outcomes of aging.

## Supporting information

**S1 File. Supplementary information.** Additional explanatory information to support the methods (sections A-B) and discussion (section C) of this study.
(DOCX)

## Acknowledgments

The authors wish to thank Mr. Yotam Hagur-Bahat for technical support.

## Author Contributions

**Conceptualization:** Sharon Gilaie-Dotan, Meir Plotnik.

**Data curation:** Amit Benady, Sean Zadik.

**Formal analysis:** Amit Benady, Sean Zadik.

**Funding acquisition:** Sharon Gilaie-Dotan, Meir Plotnik.

**Investigation:** Sharon Gilaie-Dotan, Meir Plotnik.

**Methodology:** Amit Benady, Sean Zadik, Sharon Gilaie-Dotan, Meir Plotnik.

**Software:** Amit Benady.

**Supervision:** Sharon Gilaie-Dotan, Meir Plotnik.

**Validation:** Sharon Gilaie-Dotan, Meir Plotnik.

**Visualization:** Amit Benady.

**Writing – original draft:** Amit Benady.

**Writing – review & editing:** Adi Lustig, Sharon Gilaie-Dotan, Meir Plotnik.

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
