## [Decision Letter · Decision Letter 0]

6 Aug 2024

PONE-D-24-13928The effect of virtual visual scene inclination transitions on gait modulation in healthy older versus young adults - a virtual reality studyPLOS ONE

Dear Dr. Plotnik,

Thank you for submitting your manuscript to PLOS ONE. After careful consideration, we feel that it has merit but does not fully meet PLOS ONE’s publication criteria as it currently stands. Therefore, we invite you to submit a revised version of the manuscript that addresses the points raised during the review process.

**ACADEMIC EDITOR: **The authors present an intriguing study on a research topic that will gain increasing significance in society in the future. While the rationale may be considered valid, experts in the field have identified significant flaws that must be resolved before the manuscript can be published.

We look forward to receiving your revised manuscript.

Kind regards,

Andrea Tigrini, Ph.D.

Academic Editor

PLOS ONE

Journal Requirements:

**Additional Editor Comments:**

The authors present an intriguing study on a research topic that will gain increasing significance in society in the future. While the rationale may be considered valid, experts in the field have identified significant flaws that must be resolved before the manuscript can be published.

Reviewers' comments:

Reviewer's Responses to Questions

**Comments to the Author**

1. Is the manuscript technically sound, and do the data support the conclusions?

Reviewer #1: Yes

Reviewer #2: Yes

2. Has the statistical analysis been performed appropriately and rigorously? 

Reviewer #1: Yes

Reviewer #2: Yes

3. Have the authors made all data underlying the findings in their manuscript fully available?

Reviewer #1: No

Reviewer #2: Yes

4. Is the manuscript presented in an intelligible fashion and written in standard English?

Reviewer #1: Yes

Reviewer #2: Yes

5. Review Comments to the Author

Reviewer #1: This study examined if the influence of visual cues on walking speed modulations in healthy older adults was similar to younger adults. Using a fully immersive VR system with a self-paced treadmill and projected visual scene, the inclination of both the visual scene and treadmill were manipulated independently and were found to result in similar responses. Significant correlations between the magnitude of speed modulation and visual field dependency were seen in both young and older adults, consistent with a reliance on visual system for modulation of gait in adults. While interesting, a few questions remain.

Introduction

1. What is the impact of visual acuity and sensorimotor processing on the visual field dependency in older adults? Discussing the relevant literature may help provide additional context for the stated hypotheses.

Methods

2. Was there an a-priori justification for the sample size selected? If not, please be sure to include effect sizes for all outcome measures to evaluate if study was sufficiently powered to detect changes.

3. How were physical and visual restrictions assessed? Were there visual testing performed, or were self-reported evaluations, or medical records used? Was color blindness assessed? Was visual acuity assessed, or need for bifocals, or corrective lenses?

4. Did the authors consider controlling for multiple comparisons, given large set of analyses?

Results

5. Results would benefit from reporting of effect sizes in test.

6. Be sure to use higher resolution or vectorized figures, as the current figures were difficult to read.

Discussion

6. Good discussion of the findings in relation to relevant literature.

Reviewer #2: Summary:

The authors have done interesting work in comparing the visual and body based walking speed adjustment procedures in both young and elderly people. Results have shown that both groups have responded in exertion or braking effects in similar manner while the elderly people tend to have more visual dependency suggesting that elderly people tend to rely on vision to modulate their walking pattern in response to inclined surface. However, few points were observed during reading the manuscript as follows:

- Authors have used the abbreviation “IMG” without introducing it. Furthermore, I think that it could be a typo and they meant MSI instead.

- “A positive ratio indicates that both WS and a free body accelerate/decelerate in the same direction (i.e., both accelerate when downhill and decelerate when uphill). A ratio further from zero suggests more gravitational influence on walking.” Could the author here confirm if all the values will be positive or if there is an unmentioned interpretation of negative values?

- Equations 4 and 5 requires further clarification in describing the rationale behind and defining all the terms giving attention that I observed the usage of Wvisual-cues and Wvision while the difference between them is not clear. The same applies to Wbody-based and Wbody-based cues.

- “Statistical analyses Values are presented by their group mean values (± SE).” The term values should be more specific in referring to which values.

- “The after effect, comparing the time of return from the peak/trough…”. The after effect term is not so clarifying.

- Statistical analysis were done using t-test to compare the results, were the data tested for normality?

- Authors have to add more description of the results from Fig2 regarding the cases where the Treadmill was not Ground levelled in the section of Gait speed modulations following physical and/or virtual inclination transitions since in the current version it included only information of conditions where treadmill was in ground level mode.

- In the first part of the discussion, there is a repetition of the information regarding the results of the differences in the magnitude and timing of the braking and exertion effects between young and elderly group, however it needs to be further discussed and interpreted in expansion with respect to the results section.

- “Herein we strengthen the notion that visual field dependency increases with aging [26-27], this can be seen by (1) significant increase in the older adults group compared to the young group” Increase in what? In the visual field dependency?

- Clinical values of the findings should be more emphasized within the discussions.

6. PLOS authors have the option to publish the peer review history of their article (what does this mean?). If published, this will include your full peer review and any attached files.

Reviewer #1: No

Reviewer #2: No

---

## [Author Response · Author response to Decision Letter 0]

22 Aug 2024

Response to the Editors and Reviewers

We thank the editorial board members and the reviewers for their useful comments. We have addressed each and every one of them and incorporated needed revisions in the revised manuscript. The feedback from the reviewers was particularly helpful for strengthening our statistical analyses, description of the methodology and general readability throughout the revised manuscript and also, in clarifying several issues. Point-by-point response to each of the comments of the reviewers is provided below .

Response to comments made by reviewer #1:

Comment1: “What is the impact of visual acuity and sensorimotor processing on the visual field dependency in older adults? Discussing the relevant literature may help provide additional context for the stated hypotheses.”

Response: We thank the reviewer for pointing this out. In the revised introduction we now elaborate on the possible interaction between changes in sensorimotor processing and increased reliance on visual cues (visual dependency) in older age. With respect to visual acuity, interestingly, although known to weaken with age, we found no evidence supporting vision acuity as an influencing factor which changes or diminishes the compensatory mechanism of increased visual dependency with age (p.5, par.1).

"…This reduction in older age is due to several changes that occur with aging and involves alterations in the MSI component weights i.e., delayed and/or decreased vestibular and somatosensory influx, which in turn increase visual dependency with aging [14-16]. Interestingly, this compensatory increase of visual dependency in the MSI takes place even though visual acuity is known to deteriorate with age [17]. "

Comment2: “Was there an a-priori justification for the sample size selected? If not, please be sure to include effect sizes for all outcome measures to evaluate if study was sufficiently powered to detect changes.

Response: Thank you for raising this important issue. We now report effect size along with our revised statistical analysis results, using the rank biserial correlation, in accordance with the non-parametric approach of the Man-Whitney test which was selected following comment #6 of reviewer #2 (p.13, par.1; p.13-14,16 throughout the revised results section).

"… and results effect size was estimated using the rank biserial correlation (rrb), in accordance with the non-parametric approach."

Comment3: “How were physical and visual restrictions assessed? Were there visual testing performed, or were self-reported evaluations, or medical records used? Was color blindness assessed? Was visual acuity assessed, or need for bifocals, or corrective lenses?”

Response: Thank you for this comment which motivated us to elaborate, in the revised methods section, in what manner we evaluated physical and visual restrictions of potential participants. Before entering the study, participants were required to self-report having vision acuity 20/20 (or corrected to 20/20 by means of glasses/contact lenses) and no color blindness. Regarding their physical state, we specifically inquired about any neurological, vestibular or orthopedic conditions which may hinder their ability to walk or if there are any other movement restrictions (which were all considered as exclusion criteria). To ascertain inclusion criteria, at the beginning of the experimental session, in the habituation period, the ability of the participant to walk was verified by the examiner and subjects were asked to confirm detecting the trail and parts of the scenery of the visual arena (p.6, par. 1).

"Specifically, participants were required to self-report having vision acuity 20/20 (or corrected to 20/20 by means of glasses/contact lenses) and no color blindness. Additionally, participants were inquired about any neurological, vestibular or orthopedic conditions that may hinder their ability to walk, in addition to any other movement restrictions."

Comment4: “Did the authors consider controlling for multiple comparisons, given large set of analyses?

Response: Following the reviewer's counsel we consider the significance level ("Alpha") of our statistical analysis to be 0.025 instead of 0.05, now accounting for two outcome measures (peak amplitude and timing) (p.13, par.1).

"p-value of equal or less than 0.025 (considering two variables) indicated a significant difference."

Comment5: " Results would benefit from reporting of effect sizes in test.

Response: We agree with the reviewer, please see response to comment #2 above. We now include in the revised result section of the manuscript the rank biserial correlation for each statistical comparison reported (p.13-14,16 Throughout the revised results section)

Comment6: " Be sure to use higher resolution or vectorized figures, as the current figures were difficult to read.”

Response: Following the reviewer's suggestion, we replaced Figure 1, which was indeed in a sub-optimal resolution and reconfirmed that we are adhering to the specific formatting requirements for figures provided by the journal.

Comment7: " Good discussion of the findings in relation to relevant literature.”

Response: We thank the reviewer for this encouraging feedback.

Response to comments made by reviewer #2:

Comment1: " Authors have used the abbreviation “IMG” without introducing it. Furthermore, I think that it could be a typo and they meant MSI instead."

Response: We thank the reviewer for noticing this inconsistency. Indeed, the authors intention, in the two instances where IMG was used, was for the abbreviation MSI (Multi- Sensory Integration). We have now corrected the manuscript in these two locations. (p.11, par.2; p.18, par.2).

Comment2: " A positive ratio indicates that both WS and a free body accelerate/decelerate in the same direction (i.e., both accelerate when downhill and decelerate when uphill). A ratio further from zero suggests more gravitational influence on walking.” Could the author here confirm if all the values will be positive or if there is an unmentioned interpretation of negative values?"

Response: According to equation 1 the aforementioned ratio is the outcome of dividing the area under the curve of WS by the area under the curve of the velocity of a free body. The sign is defined in accordance with the direction of the velocity change (acceleration/deceleration). Thus, the ratio can indeed be negative if WS change is in an opposite direction to this of the velocity of a free body. An example of the conditions leading to a negative ratio can be found in the older adults group for downhill walking. While a free body in a downhill scenario accelerates uncontrollably, the older adults group demonstrated WS decrease (deceleration), peaking below the SSV. This will create a negative ratio as depicted in figure 4 (upper panel; orange dots). No unmentioned interpretation nor adjustments to negative results were carried out in this section. We include the next clarification in the revised methods section (p.10, par.4).

"A positive ratio indicates that both WS and a free body accelerate/decelerate in the same direction (i.e., both accelerate when downhill and decelerate when uphill). A negative ratio indicates that WS acceleration was in the opposite direction from this of a free body. A ratio further from zero suggests more gravitational influence on walking."

Comment3: " Equations 4 and 5 requires further clarification in describing the rationale behind and defining all the terms giving attention that I observed the usage of Wvisual-cues and Wvision while the difference between them is not clear. The same applies to Wbody-based and Wbody-based cues."

Response: We thank the reviewer for raising the unclarity regarding the formulations of equations 4 and 5. We adjusted the terminology of model parameters to be more coherent and easier to follow and further revised the descriptions in this part of the revised methods section (p.11, par. 2). 

"Given that locomotion is maintained by MSI, we used a linear weighted summation model [1] to estimate the weight of visual and body-based cues [4]. The general model is presented by the following equation:

〖WS〗_combined= W_(visual cues)*〖WS〗_(vision driven)+W_(body-based cues)*〖WS〗_(body-based driven) (2) 

WScombined is defined as the resulting integrated behavior after summing the relative part of each component. On the right side of the equation, W denotes the weight of a unimodal cue (visual or body-based) and WS refers to the walking speed which was driven solely by that cue (visual or body-based). We assume that WS in conditions TLVU (WSvision driven, up) and TLVD (WSvision driven, down) are driven exclusively by uphill and downhill visual cues, respectively. Contrariwise, we assume that WS in conditions TUVL (WSbody-based drivan, up) and TDVL (WS WSbody-based drivan, down) are driven exclusively by the corresponding body-based cues. While assuming that vision and body-based cues are the main factors influencing locomotion, the weight of each unimodal cue can be calculated using the following general equations: 

W_(visual cues)+W_(body-based cues)=1 (3) 

W_(visual cues)=((〖WS〗_combined-〖WS〗_(body-based driven) ))/((〖WS〗_(vision driven)-〖WS〗_(body-based driven) ) ) (4) 

W_(body-based cues)=((〖WS〗_combined-〖WS〗_(vision driven) ))/((〖WS〗_(body-based driven)-〖WS〗_(vision driven) ) ) (5)

recall WScombined represents the congruent conditions TUVU and TDVD (i.e., combined cues) and can be calculated using the following equations for each respective inclination:

〖WS〗_(conbined,up)= W_(vision cues,up)*〖WS〗_(vision driven,up)+W_(body-based cues,up )*〖WS〗_(body-based driven,up) (6)

〖WS〗_(conbined,down)= W_(vision cues,down)*〖WS〗_(vision driven,down)+W_(body-based cues,down )*〖WS〗_(body-based driven,down) (7)

Notably, to avoid repetition of mathematical factors ('using data to explain related data'), the predicted behavior of uphill and downhill walking was estimated using the opposite unimodal weights i.e., both visual and body-based weights derived from downhill conditions was used for the prediction of uphill walking, and the uphill derived weights was used for downhill walking."

Comment4: " Statistical analyses Values are presented by their group mean values (± SE).” The term values should be more specific in referring to which values."

Response: We thank the reviewer for this comment. Our intention was to the values described in the following paragraph. In the revised version we omitted this sentence to avoid future confusion and kept this statement in the introducing paragraph of the results section. (p.13, par. 2).

Comment5: " The after effect, comparing the time of return from the peak/trough…”. The after effect term is not so clarifying."

Response: We thank the reviewer for this comment. The following clarification is now part of the 'statistical analysis' subsection of the revised methods section (p.13, par.1). 

"Additionally, we calculated and compared the time it took the participant to return from the peak/trough velocity to SSV."

Comment6: "Statistical analysis were done using t-test to compare the results, were the data tested for normality?."

Response: We thank the reviewer for raising this important issue. Indeed, the majority of data sets included in the statistical analysis of this study were normally distributed. However, in the older adults group several outcome measures of the time to peak, were not. Accordingly, and to assure robust statistical assessment in the revised manuscript we choose a non-parametric approach using the Man-Whitney test for group comparisons and the Spearman's correlation coefficients were computed to evaluate associations between outcome measures. The following citations were included in relevant location of the revised methods and results sections and all required correction were implemented (p.12, par.2). 

"In pre-hoc analyses, Shapiro-Wilk normality tests were conducted on the time and amplitude of peak outcome measures (in both TLVU and TLVD conditions), for each age group (n=2; young and older adults). Of the eight data sets assessed, normality tests indicated partially non-normal distributions for the older adults group. Accordingly, a non-parametric approach was chosen for the post-hoc age group comparison using the Mann-Whitney test. We than confirmed all outcome measures sphericity (i.e., equal variances of the differences between all groups) using the Levene's test (p>0.05)."

"…Spearman correlation coefficient was computed to evaluate the association between visual field dependence index and (i) the standardized response to virtual inclination and (ii) the age of participants."

Comment7: "Authors have to add more description of the results from Fig2 regarding the cases where the Treadmill was not Ground levelled in the section of Gait speed modulations following physical and/or virtual inclination transitions since in the current version it included only information of conditions where treadmill was in ground level mode".

Response: We thank the reviewer for this comment which driven us to include explanations regarding the non-level inclination results in the main document instead of the supplementary material file to strength the inclusiveness of the results section. The following is now brought in (p.14, par. 2).

"For the treadmill up conditions there was a gradual monotonic change in speed in both groups, with a more prominent effect in the older adult group (see Figure 2, first row, purple line (older adult group) is lower (i.e., slower) than the orange line (young group)). Interestingly, in the treadmill down conditions both groups showed opposite walking patterns; in the congruent condition TDVD, while the young group constantly increased their walking speed, the older adult group initially braked and decreased their speed, with a monotonic increase during the next minute after the transition. This pattern was roughly the same also in the TDVU and the TDVL conditions."

Comment8: "In the first part of the discussion, there is a repetition of the information regarding the results of the differences in the magnitude and timing of the braking and exertion effects between young and elderly group, however it needs to be further discussed and interpreted in expansion with respect to the results section."

Response: Thank you for this comment. Our intention in this initial paragraph was to summarize the study results to increase readability of the following discussion sections in which more in-depth explanations and interpretations are included. We accept the reviewer's comment suggesting this part to be partially redundant and in the revised manuscript we shorthand it. 

Comment9: "Herein we strengthen the notion that visual field dependency increases with aging [26-27], this can be seen by (1) significant increase in the older adults group compared to the young group” Increase in what? In the visual field dependency?"

Response: Thank you for this comment. Our intention was indeed to describe the increased visual dependency of the older adults group compared to the young, which is reflected both through a higher rod and frame index and by a positive correlation to age within this age group. We rephrased the explanation in this section in the revised manuscript (p.20, par. 2).

"Herein we strengthen the notion that visual field dependency increases with aging [26-27], this can be seen by (1) significant increase of visual field dependency as reflected by the 'road and frame' test index, in the older adults group compared to the young group, and (2) only within the older adults, there is a significant positive correlation between age and rod and frame index."

Comment10: "Clinical values of the findings should be more emphasized within the discussions."

Response: We thank the reviewer for this suggestion which we implement in the revised discussion section. The clinical values of this study's findings are now emphasized and delivered in a more concise and clear way. (p.22, par.1).

"Overall, senses and body functions are known to deteriorate with aging, such as vision [35], joint mobility [36], muscle force [37] and balance [16

---

## [Decision Letter · Decision Letter 1]

17 Sep 2024

The effect of virtual visual scene inclination transitions on gait modulation in healthy older versus young adults - a virtual reality study

PONE-D-24-13928R1

Dear Dr. Plotnik,

We’re pleased to inform you that your manuscript has been judged scientifically suitable for publication and will be formally accepted for publication once it meets all outstanding technical requirements.

Kind regards,

Andrea Tigrini, Ph.D.

Academic Editor

PLOS ONE

Additional Editor Comments (optional):

The manuscript has been carefully revised and I endorse publication.

Reviewers' comments:

Reviewer's Responses to Questions

**Comments to the Author**

1. If the authors have adequately addressed your comments raised in a previous round of review and you feel that this manuscript is now acceptable for publication, you may indicate that here to bypass the “Comments to the Author” section, enter your conflict of interest statement in the “Confidential to Editor” section, and submit your "Accept" recommendation.

Reviewer #1: All comments have been addressed

Reviewer #2: All comments have been addressed

2. Is the manuscript technically sound, and do the data support the conclusions?

Reviewer #1: Yes

Reviewer #2: Yes

3. Has the statistical analysis been performed appropriately and rigorously? 

Reviewer #1: Yes

Reviewer #2: Yes

4. Have the authors made all data underlying the findings in their manuscript fully available?

Reviewer #1: Yes

Reviewer #2: Yes

5. Is the manuscript presented in an intelligible fashion and written in standard English?

Reviewer #1: Yes

Reviewer #2: Yes

6. Review Comments to the Author

Reviewer #1: The authors are commended for their responses and have more than adequately addressed all prior reviewer feedback.

Reviewer #2: (No Response)

7. PLOS authors have the option to publish the peer review history of their article (what does this mean?). If published, this will include your full peer review and any attached files.

Reviewer #1: No

Reviewer #2: No

---

## [Editor Report · Acceptance letter]

16 Oct 2024

PONE-D-24-13928R1 

PLOS ONE

Dear Dr. Plotnik, 

I'm pleased to inform you that your manuscript has been deemed suitable for publication in PLOS ONE. Congratulations! Your manuscript is now being handed over to our production team.

Kind regards, 

on behalf of

Dr. Andrea Tigrini 

Academic Editor

PLOS ONE